# Morphology of Biomaterials Affect O-Glycosylation of HUVECs

**DOI:** 10.3390/jfb13040235

**Published:** 2022-11-11

**Authors:** Xingyou Hu, Jiaoyue Sheng, Guoping Guan, Tongzhong Ju, David F. Smith, Lu Wang

**Affiliations:** 1College of Textiles & Clothing, Qingdao University, Qingdao 266000, China; 2Department of Technical Textiles, College of Textiles, Donghua University, Shanghai 201620, China; 3Department of Biochemistry, Emory University School of Medicine, Atlanta, GA 30322, USA; 4Qingdao No. 6 People’s Hospital, Qingdao 266000, China

**Keywords:** biomedical materials, plasma treatment, cell–cell reaction, surface morphology, ICAM-1, glycosylation

## Abstract

Biomaterials have been widely used as substitutes for diseased tissue in surgery and have gained great success and attention. At present, the biocompatibility of biomaterials such as PET woven fabrics is often evaluated both in vitro and in vivo. However, the current experimental methods cannot reveal the relationship between material surfaces and cell adhesion, and few research works have focused on the mechanisms of how the surface morphology of biomaterials affects cell adhesion and proliferation. Thus, it is meaningful to find out how the altered surfaces could affect cell adhesion and growth. In this study, we employed Ar low-temperature plasma treatment technology to create nano-grooves on the warp yarn of PET woven fabrics and seeded human umbellar vein endothelial cells (HUVEC) on these fabrics. We then assessed the O-glycan and N-glycan profiles of the cells grown on different structures of the polyester woven fabrics. The result showed that the surface morphology of polyester woven fabrics could affect the O-glycan profile but not the N-glycan profile of cultured HUVEC. Taken together, the study describes the effects of the surface morphology of biomaterial on the biosynthesis of cellular glycans and may provide new insights into the design and manufacture of biomaterials used as blood vessels based on the expression profiles of O-glycans on cultured cells.

## 1. Introduction

Woven Fabrics are highly flexible and anisotropic materials, and the surface morphology of the fabrics was affected by their woven structures. The pore sizes of homogeneous microporous polyester fiber aggregates can affect the expression of adhesion molecules and consequent cell adhesion and proliferation. If nanoscale structures can be constructed on the surface of fibers, the effects of microscopic and nanoscale structural changes of textile materials on cell adhesion and proliferation can be studied in more detail. The plasma contains a large number of electrons, ions, excited atoms and molecules, free radicals, and other active particles. These active particles can elicit physical and chemical reactions such as etching, oxidation, reduction, cracking, cross-linking, and polymerization on the surface of the material. Thus, some properties of the material surface, such as wettability, abrasion resistance, or biocompatibility, can be altered [1,2,3,4].

Low-temperature plasma treatment is a commonly used material surface modification technology. In this study, we investigated the results of plasma treatment at different powers on the surface of woven PET fabrics and the effect of different surface structures on cell growth and adhesion.

Intercellular adhesion molecules (ICAM) are involved in many biological processes in vivo, the expression of which could influence cells on the inflammatory responses [5,6,7]. Recent studies showed that the membrane-bound ICAM-1 (mICAM-1) aids in helping T-cells cross the endothelial cell layer, indicating an important role in the process of cell–cell adhesion [8,9,10]. Interestingly, another type of ICAM-1, soluble ICAM-1 (sICAM-1), which was generated from the ectodomain shedding of mICAM-1, has also been shown to play an important role in regulating cell–cell adhesion. Moreover, sICAM-1 is usually expressed at sites where inflammation and lesions are present and is associated with the activation of leukocytes [11,12]. Accordingly, a high expression of sICAM-1 may suggest a pathological condition of the organism in vivo, while it may be heralded as a poor cell–cell adhesion molecule in vitro. Thus, we can predict the cell adhesion on the surface of the biomedical materials surfaces or the potential occurrence of inflammation in vivo by testing the levels of ICAM-1 in vitro.

Glycosylation is one of the most common post-translational modifications of proteins and plays critical roles in cell adhesions. There are two major types of glycosylation, mucin-type O-glycosylation (O-glycans) and Asn-linked (N-linked) glycosylation (N-glycans). O-glycans can now be assessed by Cellular O-Glycome Reporter/Amplification (CORA) technology, where cells incubated with peracetylated benzyl-α-N-acetylgalactosamine (GalNAc-α-Benzyl) convert it to a large variety of modified O-glycan derivatives that are secreted from cells, allowing easy purification for analysis by HPLC and mass spectrometry (MS), which is the most sensitive and simplest method to detect O-glycan structure in living cells [13]. When cells acquire signals such as activation, proliferation, differentiation, and apoptosis, they will promptly initiate transcription, translation, and post-translational modifications of relevant genes. Therefore, changes in cellular glycosylation may reflect the state of cell growth. For mammals, glycans can participate in many important biological processes such as cell adhesion, cell–cell signal transduction, immune response, molecular transport, and clearance. Abnormal expression of O-glycans and N-glycans in vivo have been associated with pathological conditions or diseases. For example, in tumor cells, branching and upregulated levels of sialylation in glycans often occur. Structural changes in cellular glycans are commonly monitored using mass spectrometry techniques for analysis of permethylated glycan structures because of their high accuracy and high sensitivity [14]. The permethylation process of glycans is a chemical method that replaces all hydroxyl groups (−OH) in the glycans with methyl (−OCH_3_) and is commonly used for stabilizing sialic acid, decreasing heterogeneity, increasing sensitivity, and quantification. At present, CORA, combined with mass spectrometry technology, has been used for the detection and analysis of O-glycan structures, and this method is novel, sensitive, and easy to operate. Different from the O-glycan analysis, the N-glycans on glycoproteins can be released by the specific enzyme PNGase F, permethylated, and analyzed by mass spectrometry [15,16].

It is not known if the structure changes on biomaterials could affect the cell adhesion to the surface of the materials or influence glycosylation. In this study, we made biomaterials with different surface structures by nanostructural construction of woven plain weave fabrics and low-temperature plasma etching technique, then cultured the endothelial cells on the biomaterials and evaluated the cellular adhesion and proliferation as well as the expression of adhesion molecules. We then assessed the influence of materials structure change on glycosylation using CORA for O-glycan profiling and Lectin blots before and after in-gel PNGase F digestion for N-glycan profiling. To this end, we can establish a relationship between the cell adhesion behavior on different surfaces and the expression of glycoproteins. The value and significance of surface modification of materials can be comprehensively evaluated to reveal the regulatory mechanisms underlying surface modification affecting cell adhesion and proliferation.

## 2. Method and Material

### 2.1. Plasma Treatment of PET Fabrics

The monofilaments and multifilament were treated with low-temperature argon plasma before weaving, and the conditions are shown in Table 1. By using a scanning electron microscope (SEM, ZEISS, Jena, Germany) to observe the surface of the filaments, a suitable condition was chosen to treat the fabrics.

The PET woven fabrics were fabricated with the use of a narrow ribbon shuttle loom (ASL2000-20-E, Darong Textile Instrument Co., Ltd., Wenzhou, China), with PET monofilaments for warp (30 deniers) and multifilaments for weft (30 deniers) being used in the fabrication. The designed warp density was 850/10 cm, and the weft density was 700/10 cm.

Fabrics being washed and dried were then settled into a low-temperature plasma and treated with Argon. Four fabrics marked B–E were etched at a vacuum of 20 Pa, 5 min and power 150 W, 200 W, 250 W, and 300 W, respectively. Fabrics A was considered a control group. Meanwhile, Fourier transform infrared spectroscopy (FTIR, Thermo Fisher Scientific, Waltham, MA, USA) was employed to investigate the surface chemical characteristics of the fabrics.

Finally, fabrics were cut into circles (diameter 1.3 cm) and sterilized by autoclaving for 20 min.

### 2.2. Cell Culture

The HUVECs (PCS-100-013), purchased from American Type Culture Collection (ATCC, Manassas, VA, USA), were cultured in Endothelial Cell Growth Kit-BBE (ECGKB, ATCC, USA) with 6% fetal bovine serum (FBS, Gibco, Waltham, MA, USA) and 1% penicillin-streptomycin (Gibco, Waltham, MA, USA) at 37 °C in a 5% CO_2_ incubator. Before seeding, the cells were cultured for three generations after cryopreservation and were seeded to fabrics at concentration of 2 × 10^4^ cells/well. After that, cells were cultured for one, three, five, or seven days, with culture medium being changed every two days. The experiment was replicated 3 times.

### 2.3. Cell Proliferation Assay

Cell proliferation was assessed using the Cell Count Kit-8 (CCK-8, Keygen, Jiangsu, China). The cells were cultured for 20, 68, 116, or 164 h, then 40 μL of CCK-8 solution was added to each well, and then cells were cultured for another 4 h. Then, 100 μL of solution was aspirated from each well and quickly transferred to a 96-well plate. The optical density at 450 nm (OD 450) was measured by a microplate reader (Tecan, Austria) within fifteen minutes. This assay provided a means to assess cell proliferation.

### 2.4. ELISA Assay

At the end of the 1 and 7 days of cell culture, the culture medium in each well was transferred to a 1.5 mL Eppendorf tube and centrifuged at 3000 rpm for 10 min. The levels of sICAM-1 in the supernatant were measured with an enzyme-linked immune-sorbent assay (ELISA) kit (Sigma, St. Louis, MO, USA) using the competitive method. Supernatant (50 μL) and HRP-conjugated antibody (anti-sICAM-1-HRP) (50 μL) were added to the 96-well ELISA plates, which were incubated at 37 °C for 60 min. After five washings of the plates, color development was then performed using 50 μL solution A and solution B, followed by incubation for 10 min at 37 °C. Stop solution (50 μL) was added to each well to stop the reaction. The optical density (OD) at 450 nm was measured by a microplate reader (Tecan, Austria) within fifteen minutes. As compared with values obtained using a standard solution, levels of sICAM-1 in the supernatant after HUVECs culturing on the six PET fabrics during the different time periods of culture were calculated and reported as concentrations generated (nanograms per milliliter).

### 2.5. Immunofluorescent Staining

Fabrics with adherent cells were washed with PBS three times, and then they were fixed with 4% paraformaldehyde (4 °C, 2 h). After washing 3 times, the fabrics were blocked with 2% bovine serum albumin (BSA) for 15 min at 37 °C. The cells were stained with rabbit anti-human ICAM-1-FITC (1:100, Beijing Biosynthesis Biotechnology Co., Ltd., Beijing, China) and TRITC Phalloidin (1:100, Yeasen, Shanghai, China) sequentially. Then their nuclei were stained with DAPI (Yeasen, China) for 1 min. The mICAM-1, morphology, and distribution of cells on the surface of fabrics were then evaluated from the photographs taken under a confocal laser-scanning microscope (Leica, Wetzlar, Germany).

### 2.6. Scanning Electron Microscope (SEM)

The adhesion of HUVECs on the fabrics was observed using SEM (ZEISS, EVOLS15). Fabrics with cells were washed 3 times with PBS, followed by fixation of the cells using 400 μL paraformaldehyde (4%) for 2 h at 4 °C. Subsequently, the fabrics were washed 3 times with PBS before being dehydrated through a series of ethyl alcohol gradients (30%, 50%, 60%, 70%, 80%, 90%, 95%, and 100%). The fabrics were then dried on a lyophilizer. Prior to SEM examination, the fabrics were subjected to gold sputter for 60 s and observed at an acceleration voltage of 10 kV.

### 2.7. Analysis of Cellular O-Glycans with iCORA

Fabrics were placed into wells of a 24-well plate (marked A, *n* = 3, Figure 1), then HUVECs were seeded to each well at 2 × 10^4^ cells/well. The same number of cells were seeded into wells of another 24-well plate, marked as plate B. After cells were cultured for 24 h, the culture medium was changed, and media containing 50 μM [^12^C_6_] Benzyl-α-GalNAc(Ac)_3_ and [^13^C_6_] Benzyl-α-GalNAc(Ac)_3_ were added to plates A and B, respectively, and the cells were continued in culture for 3 days as the [^12^C_6_] and [^13^C_6_] Benzyl-α-GalNAc(Ac)_3_ was converted to [^12^C_6_]- and [^13^C_6_]-labeled O-glycans and secreted into the medium. The supernatants from the wells in plate A were mixed at 1:1 (volume:volume, V:V) with the corresponding supernatants in plate B and then transferred to 6 Centricon membrane filters (MWCO: 10 KDa) marked A-F, where F was set as a control group (untreated fabrics). After removal of protein after centrifugation, the Bn-O-glycans were purified and permethylated as previously described [13].

### 2.8. Lectin Blots to Profile the N-Glycan Structures

Five different fabrics identified as A-E in replicates of 4 were placed into 20 separate wells of a 24-wells plate, and the remaining 4 wells were set as control group (no fabric). HUVECs were seeded to each well at 50,000 cells/well, and cultured for 3 days at 37 °C in a 5% CO_2_ incubator. At the end of the third day, cell pellets were collected in 1.5 mL Ep tubes after being digested by trypsin. Along with cell lysis, SDS-PAGE, lectin blotting, and exposure, the profile of cellular N-glycans was obtained.

### 2.9. Analysis of N-Glycans on Proteins with In-Gel Digestion

The collected cell pellets were directly treated with 4× loading buffer, then SDS-PAGE under 270 mA for 10 min just after the protein went across the stacking gel. The gel was stained by the coomassie brilliant blue (CBB) for 5 min, then destained for 15 min. The gel with protein bands was excised and cut into 1 mm × 1 mm pieces and then transferred to 1.5 mL Ep tubes. After in-gel digestion with PNGase F, released N-glycans were processed following a method reported by et al. [17] and analyzed on MALDI/TOF-MS.

### 2.10. Statistical Analysis

The data are reported as the means and standard deviations. Linear regression analysis and *t*-tests were used to assess the correlations and statistical significance (*p* < 0.05) between the control and test groups.

## 3. Results

### 3.1. The Surface Morphology of PET Fabrics after Plasma Treatment

The surface morphology of monofilament and multifilament after plasma treatment was observed through an SEM. Monofilaments (Figure 2I(A–D)) showed no obvious tendency in structure changes correlated with the power after being treated for 3 min. However, with a time of up to 5 min, some parallel grooves with different widths appeared on the surface of monofilaments (Figure 2I(E–H)). Meanwhile, there appeared no structure changes on the surface of the multifilament treated for either 3 (Figure 2II(A–D)) or 5 min (Figure 2II(E–H)). In this way, the woven fabrics were treated with argon plasma under 20 Pa for 5 min while the power was set at 150 W, 200 W, 250 W, or 300 W.

Results from the FTIR (Figure 3) showed that no obvious differences were observed among these five fabrics with regard to absorbance peak. Moreover, characteristic peaks obtained from all five fabrics indicated that no new functional groups were inserted after argon low-temperature plasma treatment and high-pressure steam sterilization. Accordingly, the follow-up processing employed only altered structural parameters without any effects on their chemical properties.

After seeding and culturing cells for one day, cells could be found to adhere to both the monofilaments and the multifilament (Figure 2III). However, the cell distributions on these four fabrics were very different. On the first day, it could be clearly seen that the extracellular matrix was more or less adhered to the monofilaments depending on the surface morphology. In fact, the monofilaments were treated with argon plasma, and grooves with different widths occurred on its surface. Among the four fabrics (B–E), the grooves first appeared on B and became clear on C, and the width increased on D and then disappeared on E.

As shown in Figure 2III(A), most of cells adhered to the multifilament while a small number of cells adhered to monofilaments. Interestingly, the results from SEM of fabrics B–E showed that cells of the monofilaments could be mostly found to adhere to the grooves and the pseudopod spreading wells along the fiber.

### 3.2. The mICAM-1 Distribution on the Fabrics

After fabrics were treated with argon plasma, cells were seeded on the fabrics and cultured for seven days. The mICAM-1 was immunofluorescently stained (Figure 2IV, Green); its expression and distribution reflect the cell adhesion behavior and distribution on materials.

With grooves on monofilaments, cell–matrix was easier to adhere along fibers, and the expression of mICAM-1 on fabrics B to E was higher than that on fabrics A. From B to E (Figure 2IV), the distribution of mICAM-1 changed due to varied surface morphology. On fabric E, cells grew along the side of the monofilament, and the bright green indicated a strong and concentrated expression of mICAM-1. Meanwhile, mICAM-1 was dispersedly adhered to the side of monofilaments of fabric B, indicating cell matrix adhered tightly to the surface. This is different from fabrics A and E, in which the expression of mICAM-1 was stronger but mainly concentrated where two or more cells attached. Moreover, the distribution of green fluorescence showed a cell adhesion both on the side and on the surface of monofilaments and multifilaments.

Taken together from Figure 1III,IV, the data indicates that the nanostructures of filament surface affect HUVECs adhesion through influencing expression and distribution of adhesion molecules, such as mICAM-1.

### 3.3. Changes in O-Glycan Profiles of HUVECs Cultured on Five Different Fabrics

To see whether there were changes in protein O-glycosylation in HUVECs cultured on different fabrics, we identified the possible O-glycan structures that cells made with CORA. A control group was set to normalize the intensity of [^12^C_6_] or [^13^C_6_] Bn-O-glycans using MALDI-TOF when cells were metabolically labeled with the same concentration of [^12^C_6]_ or [^13^C_6_] Bn-α-GalNAc(Ac)_3_. The result in Figure 4A showed that the intensity of peak 955 corresponds to [^12^C_6_] Bn-sialyl Core 1 O-glycans and is 80% of peak 961, which corresponds to [^13^C_6_] Bn-sialyl Core 1 O-glycans, while other four-doublet peaks (*m*/*z*: 1002 and 1008; 1316 and 1322; 1404 and 1410; 1766 and 1772) showed the same tendency. Therefore, the intensity ratio of [^12^C_6]_- to [^13^C_6_] Bn-O-glycans (with 6 Da difference) at 80% was considered as the equal abundance of [^12^C_6]_- and [^13^C_6_] Bn-O-glycans.

Thus, the intensity ratio between two peaks at 6 Da difference, [^12^C_6]_ or [^13^C_6_] Bn-O-glycans, which represent the same composition or structure of O-glycans in group A–E (Figure 5A), had been corrected by 0.8. The ratio of the peak intensity of fabrics A–E was lower than the control group that was cultured in wells without fabrics, where the peak intensity of fabric A was the lowest weakest. Compared to five experimental groups, the ratio of the peak intensity of fabric B showed no significance with fabric A on 955/961, 1317/1323, and 1766/1772 (*p* > 0.05), indicating similar expression of O-glycan on HUVECs among these fabrics. Meanwhile, the ratio of the peak intensity of fabrics C, D, and E are much higher than fabrics A and B (*p* < 0.01), and the ratio between these three fabrics showed no difference (*p* > 0.05) on peaks 955/961, 1003/1009, 1317/1323.

### 3.4. No Obvious Change in N-Glycans of HUVECs Cultured on Fabrics A–E

The multiplicity of N-glycan depends on its three kinds of structure. To investigate the N-glycosylation status of HUVECs cultured on different fabrics, we performed Lectin blotting in combination with endo-glycosidases treatments. Con A recognizes and binds to the high-mannose, hybrid, and bi-antennary N-glycans. Figure 5E shows that the cell extracts lost signal after PNGase F or Endo Hf treatment and demonstrates that the high mannose and hybrid structure of N-glycans were on the HUVECs.

To investigate the N-glycosylation of HUVECs cultured on different biomaterials, the MALDI-TOF-MS was used to analyze N-glycans on glycoproteins released by PNGase F treatment. As shown in Figure 6, *m*/*z* values of 1729, 1933, 2137, or 2341 have possible structures of *m*/*z* from the cells adhered to fabrics A–E as well as the control group. The relative intensity of each peak (P*_i_*) to the sum of the peaks intensities of all four peaks was calculated by the following formula, where I is for the intensity of each peak.
(1)$$P_{i}=\;\frac{I_{i}}{\sum_{0}^{i}I_{i}}\;(i=1729,\;1933,\;2137,\;\mathrm{and}\;2341)$$

The result showed that there was no significant difference in the relative intensity of each glycan between cells cultured with fabrics A–E and the control group, indicating that there were no significant changes in N-glycan structure after cells cultured on fabrics A–E.

### 3.5. Ar Plasma Treatment Could Promote the Cell Adhesion on Fabrics

After cells were cultured with materials for one, three, five, or seven days, cell proliferation was determined by measuring cell number using CCK-8, and sICAM-1 levels were measured by ELISA. Cell proliferation results revealed that a statistically significant standard step-like increase was present in all five experimental groups, including the control well [*p* < 0.05; Figure 5C]. Cell proliferation on day seven within the five fabrics was similar to the control well, yet cell proliferation on fabrics A–E were Control > E > C > B > D > A. Levels of sICAM-1 in cells cultured on fabrics E for one or seven days showed no statistically significant differences (Figure 5D, *p* > 0.05), while the levels of sICAM-1 in cells cultured on fabrics A to be the highest among all five fabrics.

During one to seven days of culture, cell proliferation and sICAM-1 expression in all groups were negatively correlated; while sICAM-1 concentrations were maintained at low levels, cells kept proliferating under these culture conditions.

## 4. Discussion

Studies have demonstrated that the surface property of biomaterials represents a very important factor with regard to the design of biomaterials because cell adhesion is largely influenced by the surface properties of artificial biomaterials [18,19,20,21,22]. Moreover, surface properties such as pore size, porosity, permeability, and physical/chemical properties not only impact cell proliferation but also control the direction of cell growth, as demonstrated in an artificial nerve guide conduit [23,24]. A considerable amount of research has been directed toward examining the relationship between surface structure and cell adhesion using electrospinning or sculpture on films and found that slight changes in the surface can result in enormous differences in cell adhesion [25,26,27,28]. However, despite the significance of the surface structures and cell adhesion relationship, little research has focused on understanding the mechanism underlying how the surface structure influences cell adhesion.

In this study, we use argon low-temperature plasma to etch a PET woven fabric to create grooves. Then, we investigated the changes in O-glycome and N-glycome of cells that were cultured on those different biomaterials using appropriate technologies [13,15,16]. In this way, we could investigate the relationship between surface morphology, cell adhesion, and expression of cellular glycans and gain an association that cells could adhere to different physical surfaces with modified O-glycan synthesis.

To accomplish this goal, we first explored the conditions of Ar plasma changing the surface morphology of monofilaments (Table 1). The results from this initial experiment indicated that under the pressure of 20 Pa for 5 min, we could achieve the goal by changing the power (150 W, 200 W, 250 W, or 300 W). Then, the fabrics were treated with Ar for 5 min, and the surface morphology changed widely due to the different stretch power. Compared to the surface from samples B to E, the width and deepness of the grooves increased among these three fabrics while the grooves changed to be narrow and shallow with tightly parallel arranged (Figure 2III). Results obtained using FTIR (Figure 3) confirmed that the low temperature did not change the chemical structure of fabrics by only creating some parallel grooves on the surface of monofilaments.

To access the growth and adhesion status of cells on fabrics in vitro, CCK-8 and ELISA for sICAM-1 have been used to perform a collective comparison among the five fabrics [29]. In addition, the immunofluorescence assay was used to explore the distribution of mICAM-1 on the cell plasma membrane with changes in fabric structure. The results of cell proliferation experiments showed that after 24 h of co-cultivation, the proliferation of adherent cells on the surface of each material exceeded 85% of cells in the control wells, demonstrating that after the argon plasma treatment, no toxic substances were introduced and adherent cells have comparable proliferation. After seven days of in vitro culture, the cell proliferation at one, three, five, and seven days increased over time, indicating that the etched material surface was suitable for cell proliferation (Figure 5C). Comparing the cell proliferation value on the first and seventh day among all groups, although the cell proliferation on the first day was lower, there were still differences among the groups. The cell proliferation was ranked as the control group > E (300 W) > C (200 W) > D (250 W) > B (150 W) > A from high to low. Since the materials B, C, D, and E were plasma-treated, their surfaces are rougher than the untreated material A, which allows the more extracellular matrix to adhere to the surface of the materials B–E, promoting the cell–material interaction and, to a certain extent, promoting the initial adhesion of the cells. The difference in cell proliferation on day seven was similar to that on day one among five fabrics, suggesting that the grooves produced by plasma treatment only affected the initial adhesion of the cells without affecting the proliferation rate of the cells under static in vitro culture conditions. The sICAM-1 concentrations from day one to day seven were ranked control group < E (300 W) < C (200 W) < D (250 W) < B (150 W) < A from low to high. According to other studies, the higher the concentration of sICAM-1 at the same cell concentration, the less suitable the environment is for cell adhesion. Therefore, the results of this experiment are similar to other studies. When cell proliferation is high, the concentration of sICAM-1 in the same well is low. It can be inferred that, on the surface of five different structures, the degree of ease of cell adhesion and proliferation is E (300 W) > C (200 W) > D (250 W) > B (150 W) > A.

The functions of O-glycans on glycoproteins are determined by their structures. O-glycans can protect the membrane proteins, maintain protein structure, and control active epitopes or antigenicity. O-glycans on glycoproteins can also directly participate in cell adhesion, proliferation, embryogenesis, cell differentiation, and apoptosis by controlling the expression of cell surface receptors and determining the function of receptors [30,31]. Therefore, by examining structural changes in the expression of O-glycans, we can indirectly investigate the adhesion of endothelial cells on the surface of materials. Protein O-glycosylation plays key roles in many biological processes, but the repertoire of O-glycan synthesized by cells is difficult to determine. This was made possible for the first time by the development of the CORA technology, as described by Kudelka et al. [13,32]. CORA involves metabolically labeling cells with Benzyl-α-GalNAc (Bn-α-GalNAc) to profile the cellular O-glycome with only a few numbers of cells. If the regular Bn-α-GalNAc, [^12^C_6_] Bn-α-GalNAc, and its stable isotopic compound [^13^C] Bn-α-GalNAc are used to label O-glycans of the same cells cultured at two different conditions, respectively, the same O-glycan will appear as a doublet with a difference of 6 Da in mass (*m*/*z*) if they are mixed and analyzed on mass spectrometry (MS), such as MALDI-TOF-MS. By comparing the densities of two peaks, the quantitative changes of O-glycan can be determined. In this experiment, a control group was set up, and the same concentration of cells was seeded in a 24-well plate with or without materials. [^12^C_6_] Bn-α-GalNAc in media was added to 4 wells, while [^13^C_6_] Bn-α-GalNAc in media was added to the other four wells and cultured for another three days. The collected medium was mixed at a volume ratio of 1:1, and the resulting [^12^C] − and [^13^C] Bn-O-glycans were purified and permethylated. The MALDI-TOF results are shown in Figure 4A. After calculation, the ratio of [^12^C] − and [^13^C] Bn-O-glycan peak intensities was not 1:1 after the same concentration of the compound was added to the 24-wells plate, possibly due to the error in the concentration of the compound [^12^C] − and [^13^C] Bn-α-GalNAc. Thus, the ratio of the peak intensity (^12^C:^13^C Bn-O-glycan) of the control group was set to 1, and the experimental results were corrected, which are shown in Figure 4A.

The MALDI-TOF results of the experimental group showed that the peak intensity ratio (^12^C:^13^C) of O-glycan was less than 1, indicating that the existence of materials affected the expression of O-glycan. However, compared with material A, which was not treated with plasma treatment, the peak intensity ratios (^12^C:^13^C) of all O-glycans were increased in groups B (150 W), C (200 W), D (250 W), or E (300 W), respectively, especially in group C, D, or E to 80% on O-glycan 955/961 and 90% on O-glycan 1003/1009, indicating that the plasma treatment impacts the expression of O-glycans compared to the untreated material surface. Despite this, comparing the expression of O-glycan in groups B (150 W), C (200 W), D (250 W), or E (300 W), the peak intensity ratio in the B(150 W) group is still significantly lower than the other three groups.

The microstructure of each material observed by SEM is shown in Figure 2III. After the plasma treatment, obvious groove structures appeared on the surfaces of materials C (200 W), D (250 W), and E (300 W), while only a few irregular fine lines were found on the surface of the monofilaments, and there was no regular arrangement and orientation, indicating that the existence of the groove played a certain role in promoting the expression of the O-glycans. After plasma treatment, the ratios of peaks 955/961 and 1003/1009 of samples C–D was significantly increased, especially the ratio of 1003/1009, which was similar to the control group (*p* > 0.05), the peak ratios of the other three molecular weight were still lower than the control group (100%). Based upon the glycosylation process of O-glycan (Figure 4B), among the five main peaks obtained, the intensity of peak 1317 is the highest, with an intensity ratio of 61%, indicating that the structure is the main structure of O-glycan in the selected cell line, while the 1767 peak accounted for the second (14.6%) and the minor structure. The three O-glycan structures, whose molecular weights were 1317, 1404, or 1767, all have a core 2 structure, and the chain ends contain at least one sialic acid. Therefore, the effect of material structure on the expression of O-glycan after plasma etching mainly affects the expression of Core 1 type structures and has little effect on the structure containing sialic acid and galactose, but it still improves the expression of Core 2 structures. Comparing the five structures of the experimental group, the surface treated by the power of 300 W had the most significant effect on the expression of O-glycan. After plasma treatment, the expression of the Core 1 and Core 2 O-glycans on the cell membranes adhered to the surface of the material was significantly improved.

N-glycosylation of proteins could also play a role in cell adhesion. There are three major types of N-glycans, high mannose, hybrid, or complex. To examine if there are changes in N-glycosylation of cells cultured on different biomaterials, N-glycan structures were analyzed by mass spectrometry after releasing from glycoproteins by PNGase F. As shown in Figure 6, four major N-glycan structures with *m*/*z* values of 1729 Da, 1933 Da, 2137 Da, and 2341 D were observed in cells at all culture conditions; this is consistent with the PNGase F, Endo Hf treatments, and Con A blotting results. Calculating the relative intensity of each peak to the total peaks revealed that the expression of the N-glycan structures on the cell membrane surface was not affected after co-cultivation with materials that have different microstructures (Figure 5B).

According to the results obtained by the mass-mass spectrometry, all four peaks contain the Neu5AcGalGlcNAc structure, so it can be determined that the N-glycans on the plasma membrane of the selected HUVEC are hybrid structures. The fabric surface structure does not affect the N-glycosylation.

## 5. Conclusions

Based upon the results of this study, we conclude that by changing the plasma etching power (150 W, 200 W, 250 W, and 300 W), the parallel-arranged nanoscale grooves with different depths were constructed on the warp yarn (monofilaments) of PET plain woven fabrics. Biomaterials with different structures have some impact on cell adhesion and proliferation. After fabrics were treated with argon plasma, the cell adhesion and proliferation on the surface were improved. The groove on the surface of the etching material could affect the initial adhesion of cells on the surface of the material, and the decrease in the concentration of sICAM-1 indicates that the cells are more likely to adhere and proliferate on the surface of the material. Using molecular biology methods, such as CORA, Lectin blot, and in-gel digestion, materials were analyzed to determine whether different surface structures could affect glycosylation. Meanwhile, when the etching power is higher than 200 W, the surface structure of the plasma etching material has a significant effect on the Core l structure of the O-glycan, but the effect on the Core 2 structure having a sialic acid-galactose structure is relatively weak. Interestingly, the main structure of N-glycan was hybrid, and the structure of PET fabric does not affect the structure and expression of the N-glycan. Thus, it can be concluded that the structure of fabrics could regulate cell adhesion and affect the expression of O-glycan. Moreover, further work will focus on defining a mechanistic relationship and evaluating the molecular insight into the expression glycosyltransferases affected by these biomaterials resulting in the O-glycan changes and the O-glycans on cell adhesion molecules, such as ICAM-1 on affecting their adhesion strength on the materials.

## Figures and Tables

**Figure 1 jfb-13-00235-f001:**
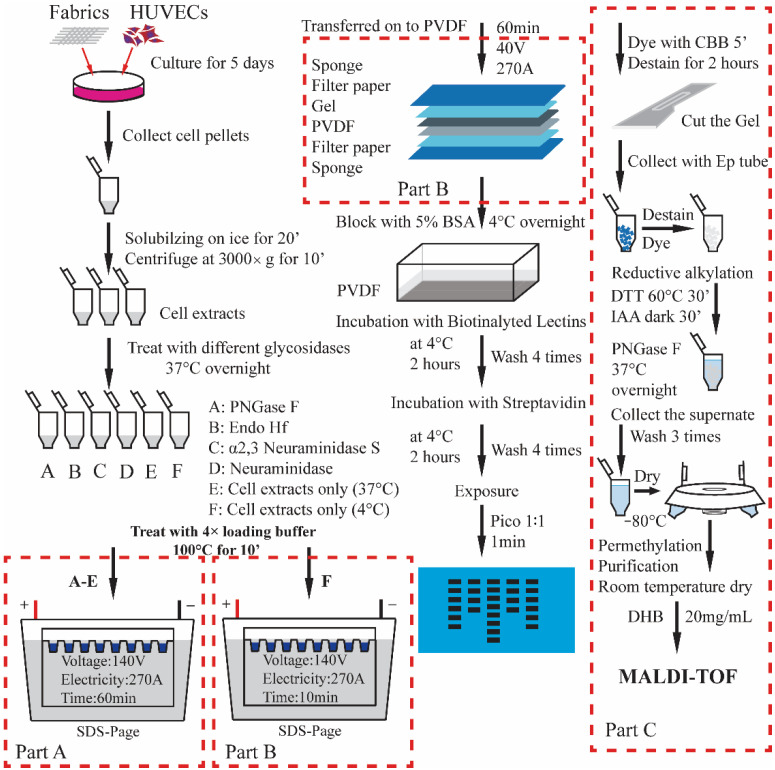
Flow chart of CORA experiment.

**Figure 2 jfb-13-00235-f002:**
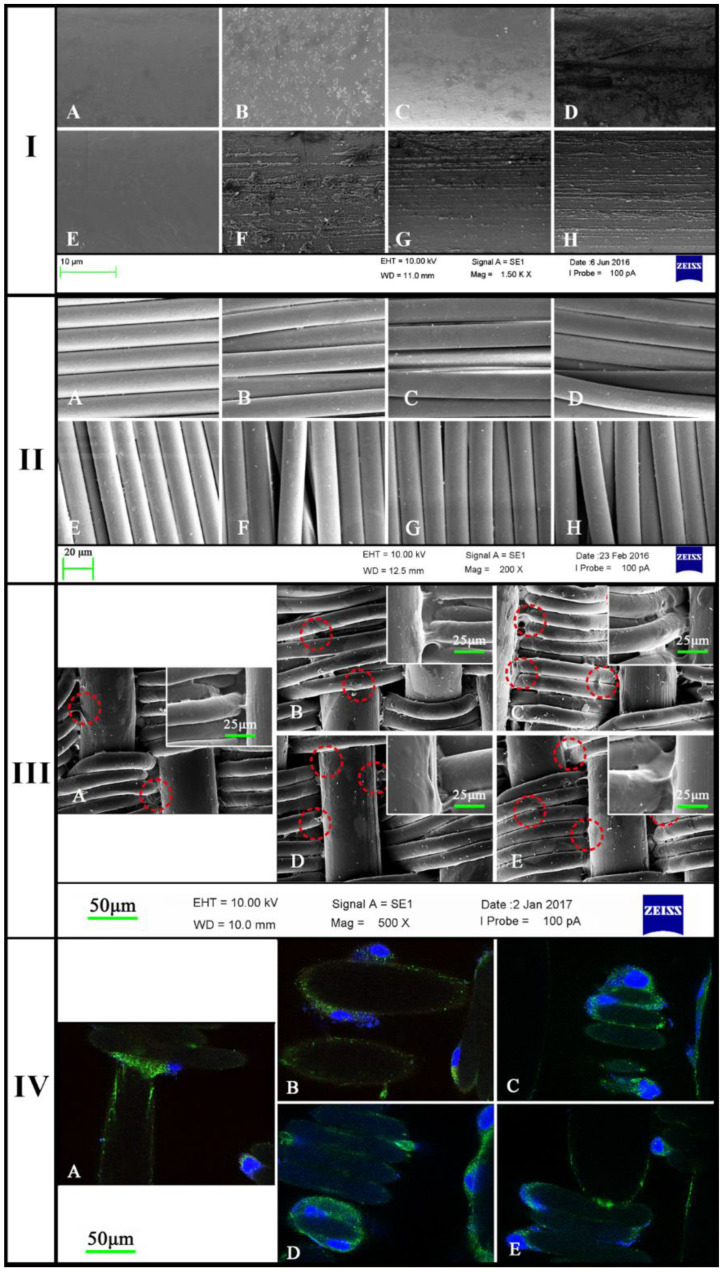
(**I**) SEM images of monofilaments treated by Ar at low-temperature plasma ((**A**–**E**): 20 Pa, 3 min, (**E**–**H**): 20 Pa, 5 min, the power was 150 W, 200 W, 250 W, or 300 W); (**II**) SEM images of multifilament treated by Ar at low-temperature plasma ((**A**–**E**): 20 Pa, 3 min, (**E**–**H**): 20 Pa, 5 min, the power was 150 W, 200 W, 250 W, or 300 W); (**III**) Scanning electron microscope (ZEISS) photograph under 500× magnification with 10 kV extra high tension (bar: 50 μm). Fabrics were treated with plasma treatment ((**A**): control group, (**B**–**E**): 20 Pa, 5 min, the power was 150 W, 200 W, 250 W, or 300 W), and cells were seeded at 20,000 cells/well and cultured for one day; (**IV**) ICAM-1 expression and distribution on plasma membranes as obtained using a laser scanning confocal microscope (LSCM) under 630× magnification with an oil immersion lens (Bar: 25 μm). Fabrics were treated with plasma treatment ((**A**): control group, (**B**–**E**): 20 Pa, 5 min, the power was 150 W, 200 W, 250 W, or 300 W). The cells were seeded at a concentration of 20,000 cells/well and cultured for seven days.

**Figure 3 jfb-13-00235-f003:**
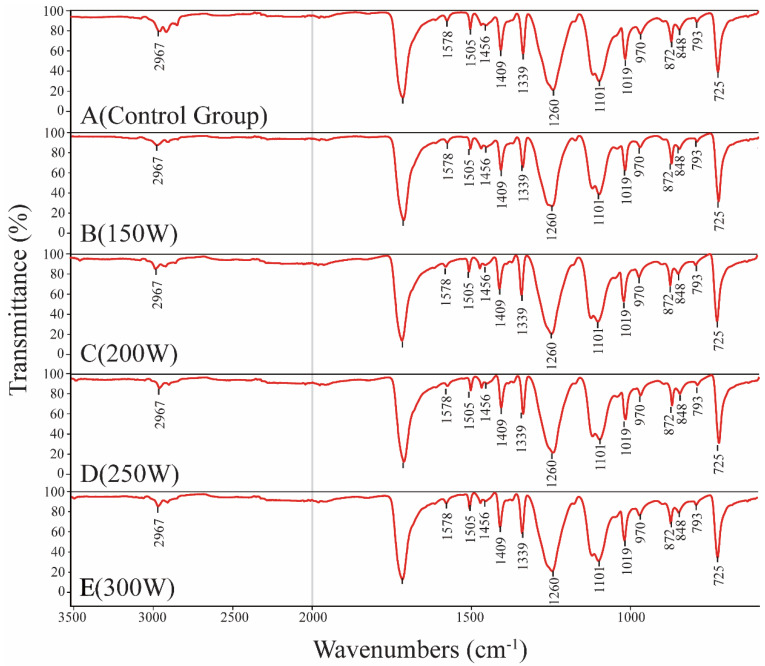
FTIR results after fabrics were treated with Ar low-temperature plasma treatment ((**A**): control group, (**B**–**E**): 20 Pa, 5 min, the power was 150 W, 200 W, 250 W, or 300 W).

**Figure 4 jfb-13-00235-f004:**
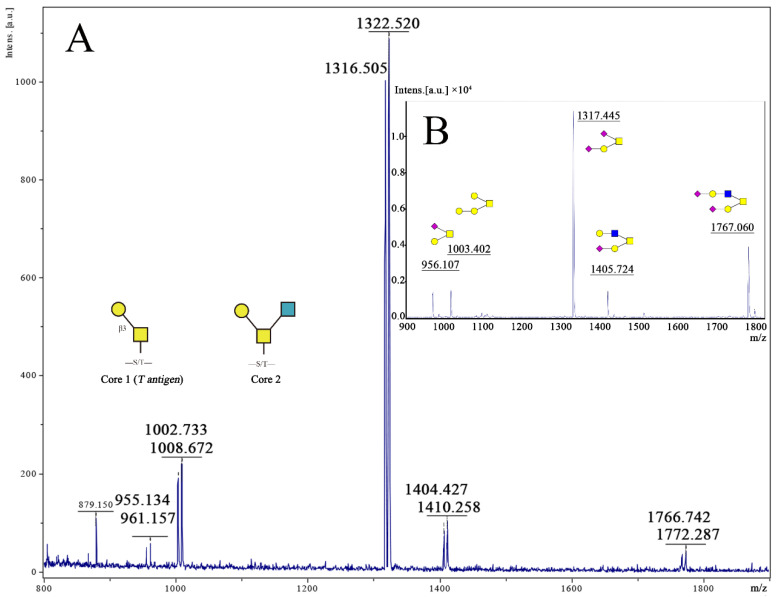
(**A**) MALDI-TOF-MS of the O-glycome of HUVECs. Cells were metabolically labeled with 50 μM of ^12^C_6_-Bn-α-GalNAc and ^13^C_6_-Bn-α-GalNAc at 1:1 ratio, and five-doublet peaks (*m*/*z*: 955 and 961; 1002 and 1008; 1316 and 1322; 1404 and 1410; 1766 and 1772) can be found. (**B**) MALDI-TOF-MS profiling of the O-glycome of HUVECs. Cells were seeded on 6-well plates, and [^12^C_6_] Benzyl-α-GalNAc(Ac)_3_ was added at a concentration of 50 μM, and the structure of each O-glycan was confirmed by MS-MS.

**Figure 5 jfb-13-00235-f005:**
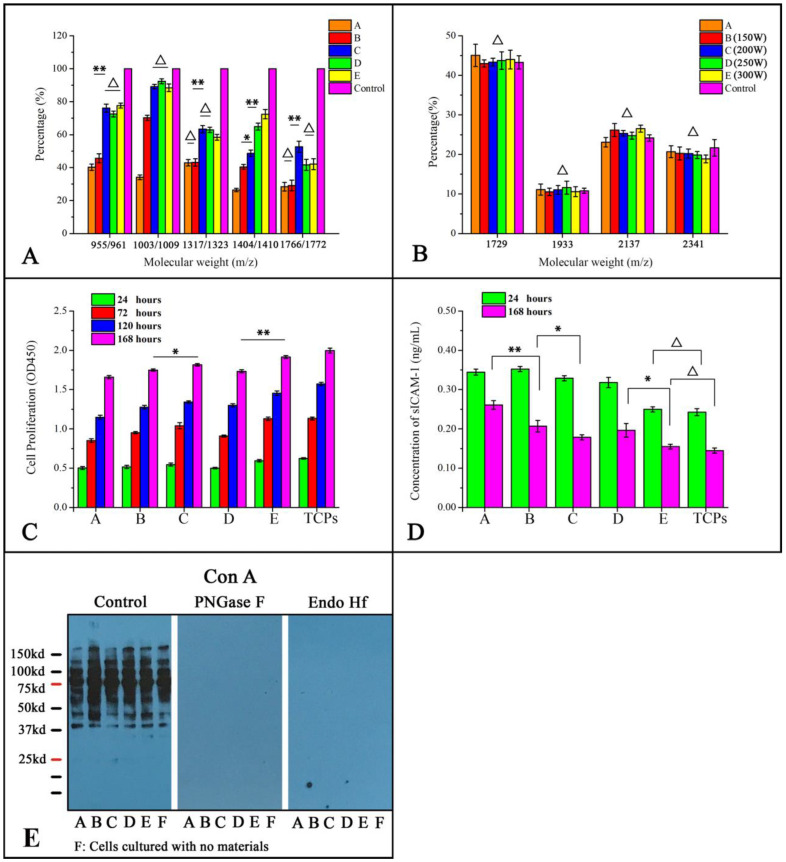
(**A**) The ratio of the peak intensity (^12^C:^13^C-glycans) of each fabric, and the control group was set to one. (**B**) The percentage of intensity of each peak to the total peaks of N-glycan. (**C**) Cell proliferation on the different fabrics after culture for one, three, five, or seven days. (**D**) sICAM-1 levels after cells are seeded on different fabrics for one and seven days. (**E**) The results of Con A lectin blots show that the cell extracts lost signal after PNGase F or Endo Hf treatment. (Sample A: control group, B–E: 20 Pa, 5 min, the power was 150 W, 200 W, 250 W, or 300 W, F: cells cultured with no materials, * *p* < 0.05, ** *p* < 0.01, △ *p* > 0.05, n = 3).

**Figure 6 jfb-13-00235-f006:**
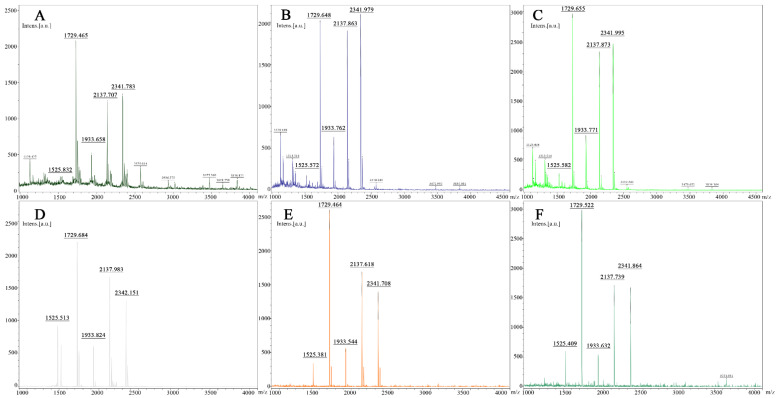
MALDI-TOF-MS/ms profiling of N-glycans of HUVECs, four major N-glycan structures with *m*/*z* values of 1729 Da, 1933 Da, 2137 Da, and 2341D were observed in cells at all culture conditions. (Sample (**A**): control group, ((**B**–**E**): 20 Pa, 5 min, the power was 150 W, 200 W, 250 W, or 300 W, (**F**): cells cultured with no materials).

**Table 1 jfb-13-00235-t001:** The parameters of plasma treatment.

Type	Vacuum (Pa)	Power (W)	Time (min)
monofilament	20	150	3
monofilament	20	200	3
monofilament	20	250	3
monofilament	20	300	3
monofilament	20	150	5
monofilament	20	200	5
monofilament	20	250	5
monofilament	20	300	5
multifilament	20	150	3
multifilament	20	200	3
multifilament	20	250	3
multifilament	20	300	3
multifilament	20	150	5
multifilament	20	200	5
multifilament	20	250	5
multifilament	20	300	5

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
