# Peer review of "Morphology of Biomaterials Affect O-Glycosylation of HUVECs"

_jfb, 2022, doi:10.3390/jfb13040235_

Round 1
Reviewer 1 Report (Previous Reviewer 1)
In this manuscript, the authors investigated the influence of the surface micro-morphology of PET woven fabrics on the HUVEC behavior. The concept and data are interesting enough. The manuscript was a resubmitted version of the original which I reviewed before.
As I checked the manuscript and the answers to my comments, I found that the authors addressed most of my comments.
Author Response
Thanks for the valuable suggestions put forward in this manuscript, which help us to improve the manuscript.
Reviewer 2 Report (New Reviewer)
Fabrication of nanostructures by simple methods has great significance for tissue engineering. The authors performed plasma treatment of different powers on PET woven fabrics to form nanostructures on the surface. The effects of these nanostructures on human umbilical vein endothelial cells were also investigated. It is an interesting work, it provides new insights into the design and fabrication of biomaterials as blood vessels. However, this paper still needs to revise many details. I suggest reconsidering the manuscript with major revisions.
1. The method of this article points out the use of argon low-temperature plasma treatment technology to create nanostructures. In contrast, the plasma treatment technology also has an impact on the properties of the material such as hydrophilicity and hydrophobicity as the author mentioned in the article. Please clarify whether the subject of the article is the effect of material properties on cells or the effect of material surface structure on cells in the abstract. Considering the main theme of the paper, please confirm the wording. If the term is wrong, please correct.
2. There have been many studies on using plasma to fabricate nanostructures and change surface properties. If the author can objectively criticize the current deficiencies in comparison with other work, the innovation of this paper will be improved.
3. The entire article lacks a schematic diagram that summarizes the workflow and meaning and uses too many words to describe it. The author can consider adding the general guidance description and a drawing of the whole work before Figure.1.
4. The cells shown in Fig.1 III are not obvious. It is recommended to add a magnified image that can clearly see the cells in each image; the fluorescence intensity in Fig.1 IV is too weak; label characteristic peaks instead of listing them all; the abscissa of Fig. 3B is not clear, so it is recommended to adjust the A and B to the same size; description of Fig 4. A and B missing from the manuscript.
5. In Figure.4D, the significance marked in the figure is difficult to distinguish whether it is a difference within a group or a difference between groups. Please mark the significance in another way. In addition, the ELISA results in this figure show that in the case of a large time difference, the difference between each group is only a few tenths of ng/ml, the difference is not obvious, please repeat the experiment again.
6. It may be worth the time to unify the font and color scheme of the figures.
Author Response
- The method of this article points out the use of argon low-temperature plasma treatment technology to create nanostructures. In contrast, the plasma treatment technology also has an impact on the properties of the material such as hydrophilicity and hydrophobicity as the author mentioned in the article. Please clarify whether the subject of the article is the effect of material properties on cells or the effect of material surface structure on cells in the abstract. Considering the main theme of the paper, please confirm the wording. If the term is wrong, please correct.
Response: Thanks for the kindly reminding. The expression in the abstract is changed from “material properties” to “the surface morphology of materials”.
In our previous study, the fabric structure has a great impact on the wettability of textile materials surface. Therefore, in this study, we selected materials with the same woven stuctures, and try to use Ar plasma to change its surface morphology, so as to try to specifically study the impact of material surface microscopic morphology changes on cell adhesion behavior as well as the glycosylation.
- There have been many studies on using plasma to fabricate nanostructures and change surface properties. If the author can objectively criticize the current deficiencies in comparison with other work, the innovation of this paper will be improved.
Response: Thanks for the suggestion. Indeed, there have been many studies on using plasma to fabricate the nanostructures on manbranes. In this study, we use the Ar plasma to treat textile materials and try to change its surface morphology is a first try, and the influence of plasma etching on the surface morphology of fabric will be studied in more detail in the future.
- The entire article lacks a schematic diagram that summarizes the workflow and meaning and uses too many words to describe it. The author can consider adding the general guidance description and a drawing of the whole work before Figure.1.
Response:The schematic diagram was added to the manuscript
- The cells shown in Fig.1 III are not obvious. It is recommended to add a magnified image that can clearly see the cells in each image; the fluorescence intensity in Fig.1 IV is too weak; label characteristic peaks instead of listing them all; the abscissa of Fig. 3B is not clear, so it is recommended to adjust the A and B to the same size; description of Fig 4. A and B missing from the manuscript.
Response: Figure 1III has been redone, and the magnified image was added to the picture, where there were cells adhered to the filaments. We aimed to express the interaction between cells through the fluorescence distribution on the cell surface, so we did not calculate the fluorescence intensity.The Figure 3B has been redrawn, and the font of ordinate has been bolded and enlarged.The description of Figure 4A and B was added in the manuscript.
- In Figure.4D, the significance marked in the figure is difficult to distinguish whether it is a difference within a group or a difference between groups. Please mark the significance in another way. In addition, the ELISA results in this figure show that in the case of a large time difference, the difference between each group is only a few tenths of ng/ml, the difference is not obvious, please repeat the experiment again.
Response:The ELISA essay was redone, and the Figure 4D was redrawn as well. Since the concentration of sICAM-1 is very low and relatively stable in the culture medium, the concentration of sICAM-1 in the culture medium will not change significantly with the increase of experimental time. Therefore, the change of the concentration of sICAM-1 in this experiment can well reflect the influence of material surface structure on cell adhesion behavior.
- It may be worth the time to unify the font and color scheme of the figures.ch.
Response:The font and color scheme were adjusted.

Round 2
Reviewer 2 Report (New Reviewer)
1. For the ELISA data in Figure 5D, the authors need to verify that they have been redone. Figure 5D in the revised version does not appear to be different from Figure 4D in the first version.
2. A suggestion that some relevant literature may help enrich the introduction section such as "An investigation on bending hysteresis of twill woven fabrics in various directions" and "Investigation on bending and creasing of woven fabrics under low curvature", etc.
Author Response
(NO.:JFB-1952735)
Article Title: Morphology of biomaterials affect O-glycosylation of HUVEC cells
- For the ELISA data in Figure 5D, the authors need to verify that they have been redone. Figure 5D in the revised version does not appear to be different from Figure 4D in the first version.
Response:I’m so sorry that the figure 5D is the old one. Revised images have been added to the manuscript.
- A suggestion that some relevant literature may help enrich the introduction section such as "An investigation on bending hysteresis of twill woven fabrics in various directions" and "Investigation on bending and creasing of woven fabrics under low curvature", etc.
Response:References have been added to the manuscript.

This manuscript is a resubmission of an earlier submission. The following is a list of the peer review reports and author responses from that submission.
Round 1
Reviewer 1 Report
In this manuscript, the authors investigated the influence of the surface micro-morphology of PET woven fabrics on the HUVEC behavior. The concept and data are interesting enough. As for my impression, I believe that additional discussion and correction will much improve the manuscript. Therefore, I think that the manuscript can be published after minor revision.
I suggest several comments as follows:
1. The title of the manuscript is non-representational. The word “Properties” makes no sense. The mainstream of the manuscript is not properties such as chemical, mechanical, and thermal properties. “Morphology” is more like, isn’t it? Please reconsider the title. (In addition, “HUVEC cells” is not adequate, because “C” of “HUVEC” means “cell”.)
2. In the Introduction section: Please spell out “CORA”, if it is an abbreviation. To make readers understand, the authors should explain “CORA” in more detail, especially on the principle.
3. In the Materials and Methods section 2.10: The authors describe “, then SDS-PAGE under 270A for 10mins”. I think that the amperage is too high. Is that correct? Usually, 270 mA or 270 V electricity are applied.
4. Figure 1, legend and Page 12 Line 346: Is the description “Ar2” correct? Usually Ar does not have covalent bond. Does it mean a special molecule of Ar at the excited state in Ar plasma?
5. Page 7 Line 214-216. The following is my purely academic question: If the chemical properties were not affected by the plasma treatment, why were the fabrics etched? Isn’t it the degradation process of the polymer molecules? Is there no change in the molecular weight of the polymer after the treatment?
6. Why did the Ar plasma work only on monofilaments while not on multifilament?
7. The authors are recommended to discuss further details what happened in the cell on the treated surface of PET filaments during the glycosylation process of O-glycan.
Reviewer 2 Report
FIrst, the premise of this paper is weak. The correlation between cell adhesion and gycolsylation is well established in the literature for many surface topographies and chemistries. So, the purpose of the paper does not reflect the appropriate scientific rigor to be published.
Second, the methods of surface modification were clearly inadequate to produce any substantial changes in the morphology, i.e., topography. The SEM evidence was presented in an inappropriate manner. Since topography was the goal, other magnifications should have been employed to enable a quantitative analysis of the surface topography of the control fibers/fabric and the modified surfaces. The comparison between the monofilaments and multifilament weaves was presented at two different magnifications and both were inadequate to evaluate any “nano” modifications intended by the Ar treatments.
Third, the differences highlighted may have been statistically significant for some of the treatments, i.e., C surface, but they were hardly biologically significant. Previous literature demonstrates that topography has a significant influence on cell adhesion and cell growth.
Finally, the long discussion appeared to be more of a review by the authors to convince the reader that the paper was of value. The result is that it left this reviewer with a negative impression of the skills of the investigators in this particular area of research. I recommend that this manuscript be rejected.